# A Scalable Heat Pump Film with Zero Energy Consumption

**DOI:** 10.3390/polym15010159

**Published:** 2022-12-29

**Authors:** Zhenghua Meng, Boyu Cao, Wei Guo, Yetao Zhong, Bin Li, Changhao Chen, Hengren Hu, Shigang Wu, Zhilin Xia

**Affiliations:** 1Hubei Key Laboratory of Advanced Technology for Automotive Components, Wuhan University of Technology, Wuhan 430070, China; 2School of Automotive Engineering, Wuhan University of Technology, Wuhan 430070, China; 3State Key Laboratory of Silicate Materials for Architectures, Wuhan University of Technology, Wuhan 430070, China; 4School of Materials Science and Engineering, Wuhan University of Technology, Wuhan 430070, China; 5School of Materials Science and Engineering, Shandong University of Technology, Zibo 255049, China

**Keywords:** net zero energy consumption, radiation cooling, thermal insulation, polyethylene

## Abstract

Radiative cooling is an effective technology with zero energy consumption to alleviate climate warming and combat the urban heat island effect. At present, researchers often use foam boxes to isolate non-radiant heat exchange between the cooler and the environment through experiments, so as to achieve maximum cooling power. In practice, however, there are challenges in setting up foam boxes on a large scale, resulting in coolers that can be cooled below ambient only under low convection conditions. Based on polymer materials and nano-zinc oxide (nano-ZnO, refractive index > 2, the peak equivalent spherical diameter 500 nm), the manufacturing process of heat pump film (HPF) was proposed. The HPF (4.1 mm thick) consists of polyethylene (PE) bubble film (heat transfer coefficient 0.04 W/m/K, 4 mm thick) and Ethylene-1-octene copolymer (POE) cured nano-ZnO (solar reflectance ≈94% at 0.075 mm thick). Covering with HPF, the object achieves 7.15 °C decreasing in normal natural environment and 3.68 °C even under certain circumstances with high surface convective heat transfer (56.9 W/m^2^/K). HPF has advantages of cooling the covered object, certain strength (1.45 Mpa), scalable manufacturing with low cost, hydrophobic characteristics (the water contact angle, 150.6°), and meeting the basic requirements of various application scenarios.

## 1. Introduction

Carbon emissions are increasing year by year, which has become a worldwide urgent environmental issue. In 2020, for example, China’s primary energy consumption reached 4.98 billion TCE, resulting in 9.9 billion tons of CO_2_ emissions [1]. Researchers focus on finding solutions from environmentally friendly materials and natural materials that are commonly used in various fields such as energy storage [2,3,4] and photocatalysis [5,6,7,8,9,10] to alleviate environmental pressures. In recent years, there have been many remarkable results, such as: Zinatloo-Ajabshir et al. for the first time using banana extract, prepared under mild conditions with good hydrogen storage performance (after 20 cycles, the discharge capacity of 2611 mA h/g) pure Pr_2_Ce_2_O_7_ nanostructures [2]. Cani et al. prepared a photocatalyst composed of C- and N-doped titanium dioxide (TiO_2_) nanoparticles that can be reused continuously [11]. These achievements promote society towards a greener, more energy-efficient direction. However, with the improvement of people’s living standards and climate change, the demand for refrigeration continues to be increasing. Approximately 1885 TWh electricity was used for space cooling in 2020, which accounted for 16% of the electricity consumption of the construction industry [12,13]. Additionally, refrigerators and air conditioners emitted CFCs that destroy the ozone layer, which causes the greenhouse effect and urban heat island effect [14]. Hence, developing climate friendly refrigeration technology draws the attention of researchers to alleviate environmental problems [15].

There is a window (8~13 μm) in the Earth’s atmosphere that cannot capture infrared light, corresponding to the peak thermal radiation spectrum of terrestrial objects near ambient temperature (~300 K). With this transparent window, objects on Earth can radiate heat into the cold outer space (3 K), thereby cutting its own temperature [16]. Based on this principle, passive radiation cooling technology develops fast and has become a research hotspot recently because the whole cooling process is zero energy consumption. Many research results show that passive radiation materials achieve self-cooling and night cooling from surrounding objects [17,18,19,20,21,22,23]. However, the greatest demand for radiation cooling is in the daytime.

It is difficult to achieve cooling under the sunset only by relying on the thermal radiation performance of material itself, when the cooling materials absorb the solar radiation heat at the same time. It is an effective way to increase the selective radiation window and reduce the solar radiation absorption at the same time. Fan et al. [24] developed HfO_2_ and SiO_2_ photonic film structures, which had 97% solar reflectivity and achieved 4.9 °C cooling in direct sunlight. Recently, cooling coatings [25], woods [26], micro-structured photonic films [27], and fabrics [28] were developed through rational material selection as well as structural design. Besides, a few novel ideas have been generated with passive radiation cooling technology, such as self-cleaning passive radiation coolers [29], self-regulating passive radiation coolers [30,31], thermoelectric [32,33], water harvesting [34,35], solar panel cooling [36,37], cooling electronic devices [38], and water cooling devices of air conditioning [39].

Passive radiate cooling materials are divided into blackbody radiation materials [40,41,42,43] and selective radiation materials according to the radiation spectrum. As selective radiation materials, TPX (contains micrometer-sized SiO_2_ spheres randomly distributed in the matrix material of polymethylpentene) [44], SiO_2_ [45], SiCNO [46], CaMoO_4_ [47], and some polymer materials [22,23] show good cooling potential when the objects placed in the circumstance without strong non-radiant heat exchange. Theoretically, if the solar absorption is suppressed and there is no non-radiative heat exchange, the cooling ability of 200 K can be achieved [48]. However, the actual non-radiative heat exchange is difficult to reduce to zero. In terms of non-radiative heat exchange, the cooling materials exchange heat with the environment through the top surface (on the environment side) and bottom surface (on the object side). Fan et al. [49] demonstrated that the sample’s maximum temperature drop reached 42 K when the selective radiator was verified in a vacuum hood. In addition, to evaluate or use the cooling capacity of radiators, the foam box and PE cover film were used in experiments shown in Refs. [50,51,52,53,54,55,56,57]. The foam box covered with aluminum foil outside blocked the ambient temperature transferred into the sample, and PE film covered the above surface of the foam box blocked the interference of ambient convection. Once the test foam box and PE film cover were removed, the radiation cooler showed poor or zero cooling capacity. Lack of temperature preservation ability of radiation cooling materials, and the side-effect of non-radiative heat absorption, seriously limits the application of passive radiation cooling materials. By enhancing the thermal resistance of the bottom surface [51,52,53,54,55,56,57,58,59,60,61] and enlarging the radiation channel of the top surface, designing a multi-layer materials structure is a worthy solution to break the application limitations of radiation cooler. Wang et al. [62] prepared ultra-high foaming rate polyethylene aerogel with high solar reflectivity (92.2%, 6 mm thickness), high infrared transmittance (79.9%, 6 mm thickness), low thermal conductivity (28 mW/m/K) and was excellently insulated. The tests showed an excited result of a temperature drop of 13 °C at noon. However, the ultra-high foaming rate polyethylene aerogel has the disadvantages of poor strength and high manufacturing cost while it is hard to large-scale manufacture.

At present, there is no excellent solution to solve the problem that radiation cooling is not ideal in practical applications, and even cannot be cooled in high convective environments. Radiation cooling technology cannot get rid of the limitation of “foam box”. This problem can be solved by isolating materials with high emissivity and high solar reflectivity. In fact, it is found that everything in nature carries out thermal radiation all the time. Using the object ‘s own radiation capacity, by covering a layer of film, namely the heat pump film (HPF), the object’s heat is like a pump to outer space with net zero energy consumption. HPF is a two-layer material, which has advantages of cooling the covered objects, certain strength, prepared efficiently with minimal cost, hydrophobic characteristics, and meeting the basic requirements of diverse application scenarios.

## 2. Cooling Principle

A schematic diagram of the HPF structure and its working principle of heat exchange is shown in Figure 1A. The HPF has two layers with the whole infrared transparent: the top thin layer of material achieves high solar reflectivity; the bottom material is well insulated.

Different with common radiate cooling emitters (thickness is <1 mm), whose cooling performance is usually analyzed by assuming that the temperatures on the upper and lower surfaces of the emitter are equal, the HPF (≈4 mm) has much higher thickness, up to the millimeter scale. There is a significant temperature difference between the upper and lower surfaces of the object, and the radiation distribution in the thickness direction needs to be considered. Light propagation is exponentially distributed along the thickness direction in an object (Figure 1B and Appendix A). HPF is infrared transparent within the atmospheric window, but still has a weak radiative power. Hence, considering the radiation distribution of each part of the material, the overall heat dissipation of the radiator is defined as (Appendix A):(1)$$Q_{net}=Q_{r}-Q_{a}-Q_{nonrad}-Q_{sun}$$
where $Q_{r}=P_{r\;ZnO}+\varepsilon_{2}P_{r\;bubble\;film}+\varepsilon_{2}\varepsilon_{3}^{n}P_{r\;object}$ is the total radiated power emitted by the sample.$\;\varepsilon_{2}$ is the IR transmittance of the ZnO film and $\varepsilon_{3}$ is the IR transmittance of the bubble film (0.8). $Q_{a}$ is the amount of incident atmospheric radiation absorbed by the sample. $Q_{nonrad}$ is the non-radiative heat absorbed by the sample, which is generated by heat exchange through environmental convection and conduction, etc. ($Q_{nonrad}=h_{c}\left(T_{amb}-T_{s}\right)$, $h_{c}$ is is the non-radiative heat transfer coefficient between the environment and the radiant cooler, $\;T_{amb}$ is ambient temperature and $T_{s}$ is the temperature of the radiant cooler), $Q_{sun}$ is the solar radiation absorbed by the sample.

The top layer is distributed Nano-powder with a high refractive index and low absorption. Selected materials with appropriate particle size can generate strong Mie scattering in the solar band without affecting the incidence in the atmospheric window band, resulting in high solar reflectivity ($Q_{sun}\;$is low) and infrared transmittance (8–13μm) simultaneously. With the increase of solar absorption depth, the cooling effect becomes worse (Appendix A). Therefore, the use of a thin layer of material to reflect sunlight can greatly reduce the absorption of solar heat. The bottom layer is a bubble film with low thermal conductivity, which can block the heat conduction between the object and the environment. The bubble film is also infrared transparent. Hence, HPF has high solar reflection, infrared transmission in the 8~13 μm wavelengths, and low thermal conductivity. In addition, sufficient interface thermal resistance induced by the non-close contact between HPF and the object will also reduce the non-radiation heat transfer between the object and the environment ($Q_{nonrad}$ is low). These conditions work together to ensure $Q_{net}$ > 0.

## 3. Materials and Methods

### 3.1. Materials

Polypropylene (PP) film (25 μm thicknesses) was manufactured by Shenzhen Rongmaoda Electronic Materials Co., Ltd. PP is infrared (8–13 μm) transparent (Appendix A) and can be used as a reflective layer substrate.

ZrO_2_, MgO, Al_2_O_3_, and ZnO nano-powders (the spherical equivalent peak diameter ≈500 nm) were manufactured by Hangzhou Hengna New Materials Co. The nano-powders are pure and free of other elements. Nano-powders with a high refractive index can undergo strong Mie scattering and achieve a high solar reflection effect. Here, ZnO Nano-powders, which have a refractive index of about 2 in solar wavelengths [63,64], also have the best IR transmission (Appendix A) and are selected to be the high solar reflection top layer. Based on Mie scattering theory, the size of ZnO particles is optimized. The normalized scattering cross section of ZnO nanoparticles was calculated (Appendix A). When the particle size is 100~1000 nm, the scattering cross section of visible and near infrared band is significantly improved, but the improvement of the atmospheric window band is not obvious. Reasonably, ZnO particles with the spherical equivalent diameter of 500 nm are selected (Appendix A). Besides, ZnO Nano-powders have an extinction coefficient being close to 0 in the 0.3~15 μm band, which means that the light absorption of the particles in these bands is very weak. Hence, the prepared ZnO film has nearly zero solar absorption, high solar band reflectivity, and a certain infrared (8~13 μm) transmission performance.

Polyethylene bubble film (PE bubble film, 2 mm thickness) was manufactured by Jiangmen Henghou Plastic Products Co. When PE films enclose air, the thermal conductivity of bubble film is only 0.04 W/m/K. The molecular structure of PE is composed of C-H and C-C bonds, which has no obvious absorption peak in the range of 8~13 μm brands. Hence, infrared transparent air and PE ensure the high transmittance of the bubble film in the atmospheric window (Figure 2A and Appendix A).

Ethylene-1-octene copolymer (POE 7447) were manufactured by Dow Co. POE is often used as a toughening agent in modified PE or PP products. POE is infrared transparent in the atmospheric window band (Appendix A).

Petroleum ether (the boiling point: 60–90 °C) was manufactured by Macklin.

Polydimethylsiloxane (PDMS) Sylgard 184 was purchased from Dow Corning and is the ideal radiation cooler [65]. It comes as a two-part elastomer kit (the pre-polymer and curing agent).

### 3.2. Methods

Outdoor Test Platform: The test samples were placed on a PE foam box wrapped in aluminum foil. Aluminum foil reflects most of the sunlight and the foam box has low thermal conductivity to prevent external temperature intrusion into the radiation cooler from the bottom. The ambient temperature was measured by a Stevenson screen, and the temperature measurement data were exported by multi-channel temperature logger. Anemometer was used to measure the ambient humidity and temperature, and a Pyranometer was used to measure the solar radiation intensity.

Characteristic of Spectral Performance: The reflectance of ZnO films was measured with a UV-Vis NIR spectrophotometer (Lambda 750 S 200–25,000 nm) with an integrating sphere, and the IR transmittance of ZnO films, bubble film, and HPF were measured with a Nicolet iS50 (Thermo Fisher, Waltham, MA, USA). The IR transmittance of building materials, PDMS, PE, and PP films were measured with a NEXUS Fourier transform infrared spectrometer (Thermo Fisher, Waltham, MA, USA). Part of the IR photographs was taken with an IR gun.

Characteristic of Microscopic Morphology: Photographs of the particle distribution of ZnO film layers were taken by field emission scanning electron microscopy attached to a X-Max N80 energy spectrometer (JSM-7500F, JPN). The size of ZnO particles was tested by a laser particle sizer (Malvern ms3000, GBR).

Mechanical test: HPF tensile strength was tested by the Instron 5967 universal material testing machine.

### 3.3. HPF Preparation

PE bubble film is manufactured by the conventional large-scale extrusion blow molding process (Figure 2A). Under the condition of 50 °C water bath heating, 2.5 g POE7447 particles were dissolved in 50 mL petroleum ether to form POE glue. POE glue can be stable at room temperature and be beneficial to the subsequent process of ZnO powder. Then, 20 mL POE glue was combined with 10 g nano-ZnO particles and stirred for 2 h. The uniformly mixed solution is coated onto the PP substrate by a film scraping machine. ZnO film is formed after petroleum ether evaporation. ZnO film and PE bubble film are stuck together with POE clue to prepare the HPF (Figure 2A,B). The whole process of preparing HPF requires simple equipment, short preparation cycle, and even no high temperature environment. All materials used are easy to obtain and are cheap (Appendix A).

## 4. Results and Discuss

### 4.1. Optical and Thermal Insulation Properties of HPF

To evaluate the effect of different ZnO film thicknesses and bubble film layers on the thermal performance, based on the optical test of HPF (Figure 3A,B), the calculation model of HPF was established (see Appendix A). The effects of thermal resistance and thermal radiation intensity on cooling performance were calculated by varying the number of bubble film layers. The effects of solar reflectance and thermal radiation intensity on cooling performance were calculated by varying the ZnO film thickness. The calculation results are shown in Figure 3C,D.

For the number of bubble film layers, it can be seen that one-layer or two-layer bubble film is the preferred choice for the cooling effect. The one-layer infrared transmittance of it in the atmospheric window is about 80% (Figure 2A). The thermal conductivity of bubble film is 0.04 W/m/K. The cooling capability of the sample with two-layer bubble film is already close to the optimal value that can be achieved.

For the thickness of ZnO, the infrared transmittance of the ZnO film decreases and the reflectance increases as the thickness increases in a range. Hence, the cooling performance of the HPF increases with the increase of ZnO thickness when solar radiates severely during the daytime. When it is greater than 75 µm, however, the thickness has little effect on the reflectance, while it has a side effect in IR transmittance. It will not be useful for the cooling performance of HPF. Outdoor experimental results validate the calculations (see Appendix A). In design and experiments, the preferred HPFs were prepared using two-layers of bubble film and 75 μm thickness ZnO film. Infrared transmission consequences of HPFs are shown in the Appendix A.

### 4.2. Characterization of HPF

Different from previous research works [28], columnar powder instead of sphere powder was selected to make ZnO film (Figure 2B), in order to low-cost scaled manufactured with low processing requirements. Some of the columnar particles reach micron size in the lengthwise direction, and the large size span of columnar ZnO is useful for achieving high reflectivity in a wider range of solar wavelengths. Although it affects the transmission in the atmospheric window band, fairly good cooling effect can be achieved. The SEM photos of the cross section (Figure 4) show the holes left by the volatilization of petroleum ether, which are conducive to the scattering of light. The POE is homogeneously mixed with ZnO particles to form a non-flaking film.

### 4.3. Outdoor Cooling Test

#### 4.3.1. Pump Heat without Auxiliary Heat Insulation

HPFs were prepared with 75 µm ZnO film and 4 mm bubble film, and a 24 h test was performed on covering on the PDMS coated Polyethylene terephthalate (PET) film (Figure 5A), where the infrared emission test of PDMS (200 μm) is shown in Appendix A. The experiments performed on a building roof in Wuhan, China (30°31′16.62″ N, 114°20′46.32″ E) on November 2021.

HPF was directly placed on the foam block and exposure to the environment, the test devices are shown in Figure 5A. For comparison, a control specimen with a foam box was tested at the same time. The commonly used foam box can hinder the non-radiative heat gain of the radiator and the environment, so the cooling capacity of the specimen in the foam box does not represent its effect in the natural environment. By abandoning the foam box, the HPF experimental device is more in line with the actual application scenario, and the experimental results are more convincing.

The test results (Figure 5B,D,E) show that, the HPF-covered PDMS (HPF sample) could cool down an average 7.15 °C steadily within a day, and its cooling ability is still better than the control specimen (6.68 °C). Although the difference of two experimental results throughout the day is small, the HPF sample shows a more stable temperature dropping. Temperature variance calculations show (Appendix A) that the HPF sample has the smallest variance values ($S_{HPF}=6.69$, $S_{trantional\;sample}=8.3$, $S_{ambient})12.6$) compared with the ambient and the control specimen. The HPF with the low thermal conductivity (Appendix A), as the same as an air gap between the radiator and the environment created by the foam box, is also beneficial to reduce the non-radiative heat exchange between the radiator and the environment. Moreover, the HPF itself has high solar reflection performance, and advances in reducing influences by solar radiation. The foam box is easier to receive the solar radiation and warming during the daytime, resulting in a large difference in the average temperature drop of the internal radiator day and night. At the same time, a windless sub-environment was created with a transparent PE film wind shield to test the cooling of HPF. In this condition of low ambient convection, the HPF-covered PDMS object cooled down an average 11 °C during daytime (Appendix A). At the same time, the HPF outdoor experimental results are compared with the recent research (the optical property of the radiation cooler is similar), and the cooling effect is not much different (see Table 1 and Appendix A). The difference is that other studies use foam boxes to block the non-radiative heat exchange between the environment and the radiator, but the HPF test is not. HPF can be laid on a large scale on the outer surface of the building to achieve ideal cooling, while the large-scale installation of foam boxes is a challenge.

#### 4.3.2. Pump Heat Covered on Different Materials

Cooling tests on traditional building materials (such as bitumen, cement, and silicon carbide) were also conducted (Figure 5C and Appendix A). HPF can reflect sunlight greatly, so even pure black bitumen can achieve cooling below the ambient temperature when covered by HPF. Despite the different thicknesses of the three materials (Silicon carbide ≈ 200 μm, Bitumen ≈ 500 μm, Cement ≈ 2 mm), the cooling effect does not differ much (all can achieve >5 °C cooling) because their emissivity in the atmospheric window band does not differ much ($\varepsilon_{Bitumen}\approx 0.95$, $\varepsilon_{Cement}\approx 0.99$, $\varepsilon_{Silicon\;carbide}\approx 0.99$).

In fact, almost all natural objects have a high radiation capacity when they are thick enough. However, the objects cannot cool down during the day due to solar absorption. When covered by HPF, almost all materials tend to cool down to a certain extent, depending on their own properties (see Appendix A). It is believed that HPF can satisfy most ordinary objects for providing heat pump cooling.

#### 4.3.3. Pump Heat under Strong Convection Environment

The HPF cooling performance under a high convection environment was tested. A PDMS object covered by HPF was fixed steadily on the roof of a driving car with an average speed of 70 km/h. The surface heat transfer coefficient ($h_{s}$) of the object is about 56.9 W/m^2^/k. The temperature change test results show that, the object cooled down an average 3.5 °C in a 40 minute journey (Appendix A).

In Formula 1,$\;Q_{nonrad}=h_{c}\left(T_{amb}-T_{s}\right)$. For radiant coolers,$\;h_{c}$ satisfies the empirical formula [66]: 1. For a set-up without a wind shield, $h_{c1}=8.3+2.5v$; 2. For a set-up with a wind shield, $h_{c2}=2.5+2v$; 3. In a closed chamber, $h_{c3}$ = 2.5 W/m^2^K. *v* is the wind speed, m/s. For the HPF covered radiant coolers, $h_{c}=\frac{1}{R+(1/h_{c1)}}$, *R* is the thermal resistance of the HPF, R= 0.1 m^2^K/W. Based on the work of other researchers, Table 2 was plotted. It can be seen that, when out of the foam box, the cooling effect of the radiation cooler is much worse. The data in bold font format are calculated based on the original paper data (assuming that the non-radiative heat transfer power of different environments is the same). The radiation cooler without wind shield cools <3 °C when *v* is just set to 1 m/s. However, the cooler under HPF can still be cooled 3.5 °C even with a wind speed of 19 m/s (Speed 70 Km/h) due to its low thermal conductivity. There is little research to achieve cooling in a highly convective environment without a foam box.

To explore the effect of $h_{s}$ on cooling, the cooling effect of HPF was calculated (Figure 6A,B and Appendix A) based on Appendix A. PDMS film was used as the object to analyze the cooling effect of the HPF. $h_{s}$ is set between (10–80 W/m^2^/k), and the number of bubble film layers number is set to (0–4).

The results show that, when the $h_{s}$ is lower than 40 W/m^2^/k, the cooling ability of the HPF gradually decreases with the increase of $h_{s}$. It will basically remain stable when $h_{s}$ is larger than 40 W/m^2^/k. In this situation ($h_{s}$ > 40 W/m^2^/k), the PDMS object covered by HPF cools down more than 3.4 °C. In contrast, the PDMS object covered by ZnO film only cools down about 1 °C. Besides, different with the low $h_{s}$ situation test results, the layers number of bubble film of HPF has little influence of cool ability. It is considered that this is because the transmittance, heat insulation, and radiation performance are changed accordingly.

It shows that there is still room for performance improvement for HPF (Figure 6B). The infrared transmittance at 8~13 μm of the HPF cooler is being optimized and improved. According to deduction, HPF could cool down the covering objects to about 24 °C, if some methods are adopted to increase the transmittance of HPF to 90%, such as applying spherical ZnO particles and fabricating with a complex fine mixing process.

### 4.4. Additional Performance

As previous research [62] showed, the infrared transparent polyethylene aerogel (PEA) has better characteristics (6 mm thickness, solar reflectance 92.2%, infrared transmittance 79.9%, thermal conductivity 28 mW/m/K) and 13 °C cooling ability at noon. However, it has insufficient mechanical properties after freeze-drying preparation, which may be restricted to apply. HPF should maintain certain strength in practical application. The tensile test results showed that the tensile strength of HPF reached 1.45 Mpa, which was significantly higher than that of PEA (Figure 7A and Appendix A). HPF shows good flexibility and bendability: The nanoparticles adhere closely without peeling even if the ZnO film is folded into a cranes shape.

In addition, the water contact angle of HPF is as high as 150.6°, hence the blue ink will not leave traces on the surface of HPF. It shows that HPF owns a certain self-cleaning ability (Figure 7B,C), which meets the requirements of construction or outdoor device shield applications.

## 5. Conclusions

A “heat pump film (HPF)” scheme was proposed and a two-layer material for cooling the covered object was developed. HPF, which is made of combined ZnO film and PE bubble film by a simple scalable production process, has high solar reflection and infrared transmission characteristics. Additionally, the raw materials are easily available and low cost. Almost all common objects covered by HPF will be cooled by their own radiation to outside space with zero energy consumption. Theoretical and experimental proofs show that HPF has a favorable cooling effect: the object covered by HPF in outdoors cools down greater than 7 °C throughout the day, and there is still a cooling effect of greater than 3.5 °C under a high convection environment (56.9 W/m^2^/k).

HPF has certain strength (1.45 Mpa), good flexibility, bendability, and hydrophobic characteristics (the water contact angle, 150.6°). HPF is also easy to lay and install and is free from the limitations of the insulation box. HPF is suitable for widely applied scenarios, such as the building exterior surface, power base station cabinet, refrigerated transport vehicles cover.

The development of HPF can solve the practical application problem of radiation cooling technology to achieve ideal cooling in a high convection environment. Nevertheless, the performance of HPF continues to have much room for improvement. The infrared transmittance at 8~13 μm of the HPF cooler is being optimized and improved. According to deduction, a cooling effect of more than 20 °C can be achieved at a static environment, if some methods are adopted to increase the transmittance of HPF to 90%, such as applying spherical ZnO particles and fabricating with a complex fine mixing process. The wide application of HPF will make a contribution to global energy conservation and carbon reduction.

## Figures and Tables

**Figure 1 polymers-15-00159-f001:**
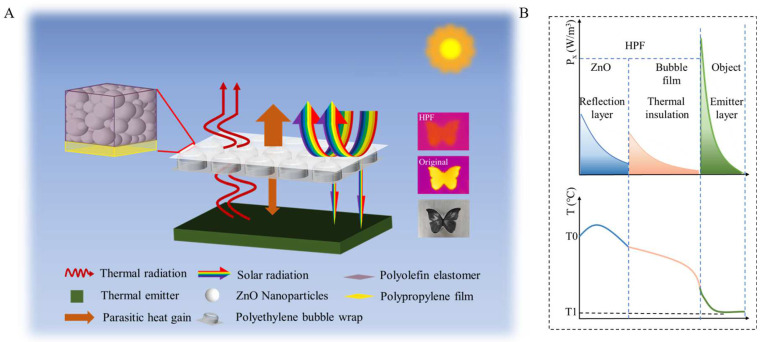
(**A**) Structural of HPF (ZnO film top layer and air bubble film bottom layer). (**B**) The cooling trend graph: The radiated power of each part of the material is distributed exponentially along the thickness direction.

**Figure 2 polymers-15-00159-f002:**
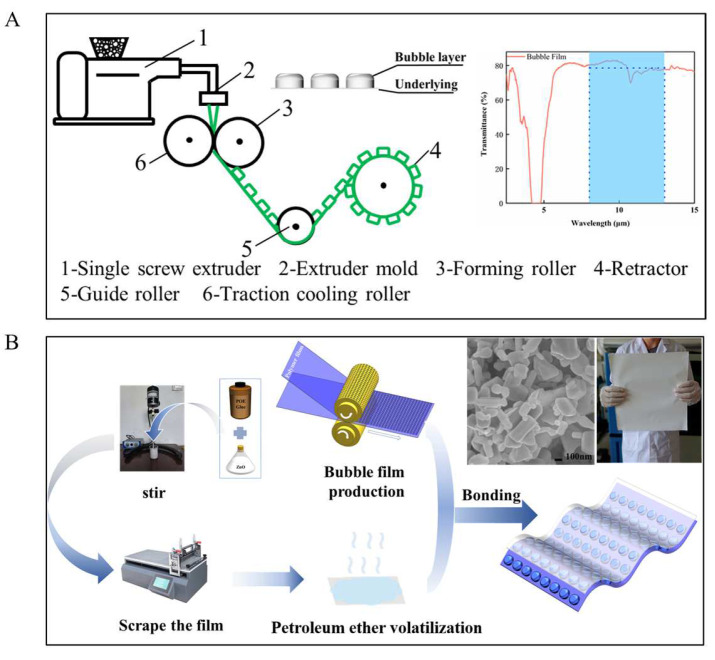
(**A**) Schematic diagram of the production process of PE bubble film, including measured transmittance of PE bubble film (2 mm thick) over MIR wavelength. (**B**) Schematic diagram of the production process of HPF, including the SEM photo of the upper surface of HPF.

**Figure 3 polymers-15-00159-f003:**
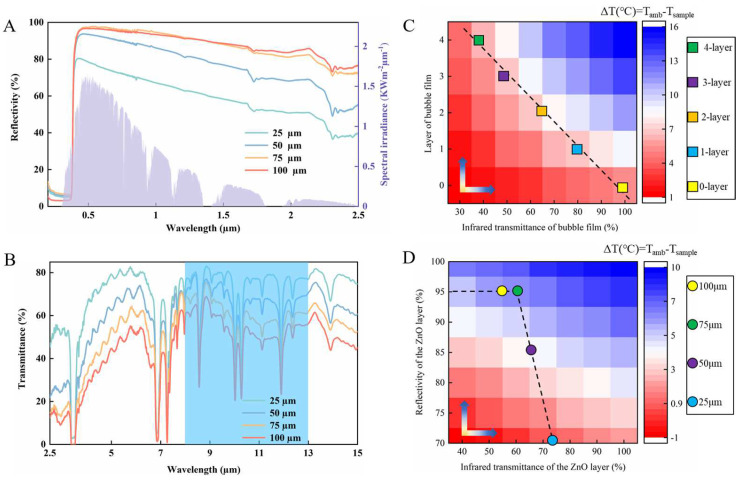
(**A**) Measured reflectance of different thicknesses of ZnO films over UV-VIS-NIR wavelengths, with the AM 1.5 solar spectrum. (**B**) Measured transmittance of different thicknesses of ZnO films over MIR wavelengths. (**C**) Model calculation: Setting the same object (PDMS, 200 μm) and ZnO (100 μm). Cooling results of object covered by 0–4 layers of bubble film. (**D**) Model calculation: Setting the same object (PDMS, 200 μm) and bubble film (2 layers). Cooling results of object covered by different thickness ZnO.

**Figure 4 polymers-15-00159-f004:**
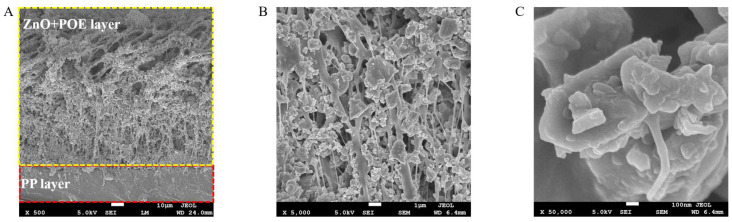
SEM photos of cross section of ZnO layer at different magnifications. (**A**) The scale is 1 mm. (**B**) The scale is 1 μm. (**C**) The scale is 100 μm.

**Figure 5 polymers-15-00159-f005:**
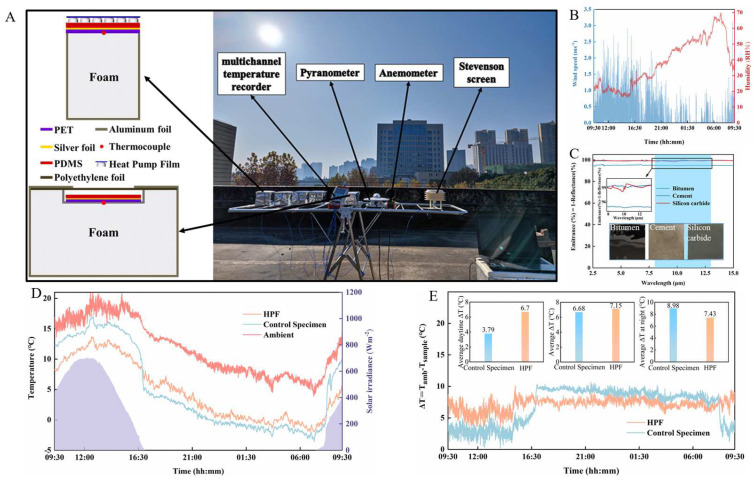
(**A**) Field test device, test location Wuhan, respectively, ambient temperature, sample temperature, wind speed, humidity, and solar radiation intensity, test device has temperature measurement device and control specimen device. (**B**) Wind speed and humidity testing during static roof testing. (**C**) Measured emissivity of building materials (bitumen, cement, and silicon carbide) over MIR wavelengths. (**D**) Sample temperature and ambient temperature testing under setup and control specimen setup, as well as real-time solar intensity testing. (**E**) Temperature variation of static roof test between device samples and control device specimen.

**Figure 6 polymers-15-00159-f006:**
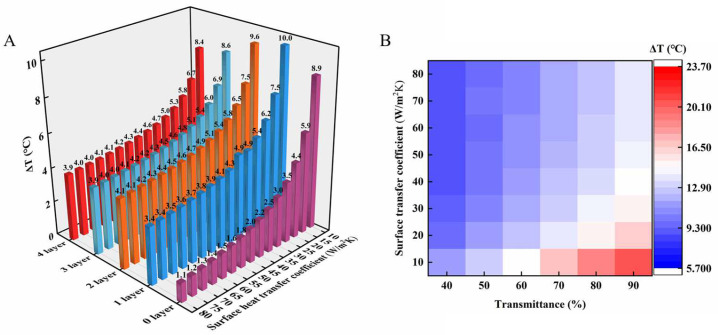
(**A**) Model calculation: cooling histogram of samples with different $h_{s}$ and different layers of bubble film insulation. (**B**) Model calculation: cooling forecast hotspot diagram with different $h_{s}$ and different transmittance (8–13 μm) of HPF.

**Figure 7 polymers-15-00159-f007:**
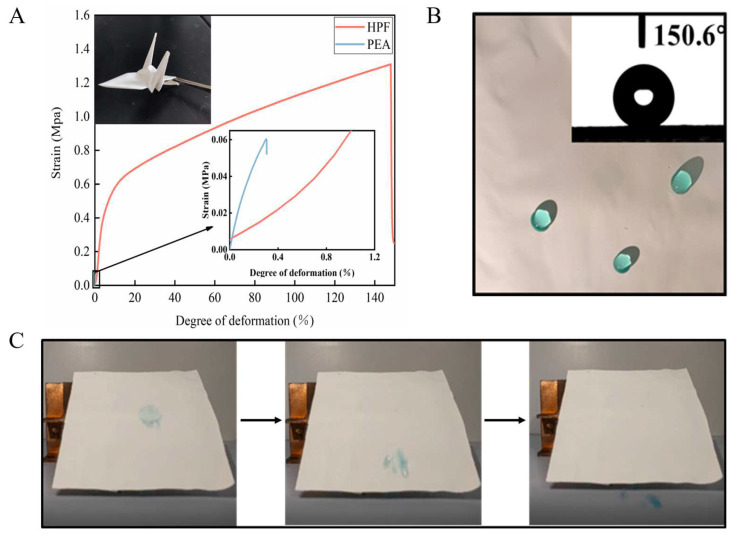
(**A**) Mechanical properties testing of PEA and HPF where the ZnO film used for the preparation of HPF can be folded into a cranes shape. (**B**) Water contact angle testing of HPF surfaces. (**C**) Demonstration of HPF’s self-cleaning effect.

**Table 1 polymers-15-00159-t001:** Comparison between different testing devices [42,51].

Serial No.	Testing Device	Materials	Reflectance (%)	Emittance (%)	$\Delta T=T_{amb}-T_{s}$
(Wu L.) In a closed chamber	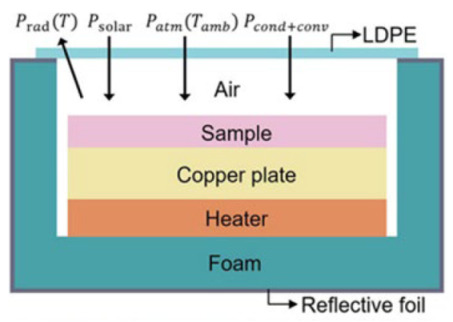	polymethyl methacrylate (PMMA)	95%	98%	6–8.9 °C
(Wang F.) with wind shield	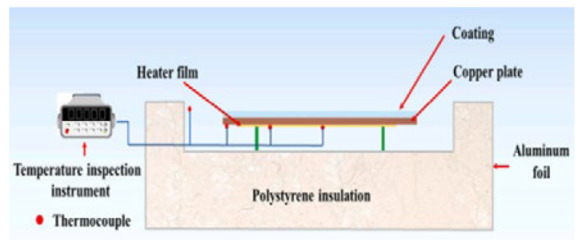	BaSO_4_ and SiO_2_	95%	96%	6.2 °C
HPF without wind shield	See Figure 4A	PDMS and HPF	≈94%	97%	7.15 °C

**Table 2 polymers-15-00159-t002:** Cooling results in different environments.

Serial No.	In a Closed Chamber	With Wind Shield	Without Wind Shield
(Wu L.)	6–8.9 °C	3.3–4.9 °C	1.4–2.1 °C
(Wang F.)	×	6.2 °C	2.6 °C
HPF	×	×	7.15 °C

## Data Availability

Not applicable.

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
