# Peer review of "A Scalable Heat Pump Film with Zero Energy Consumption"

_polymers, 2022, doi:10.3390/polym15010159_

Round 1

Reviewer 1 Report

1- The authors are recommended to avoid using pronouns in the technical writing. Thus, kindly revise all the sentences in the manuscript such as “We propose….”.

2- Please remove “and” from the start of sentence such as “. And the bubble film ….”. This is applied for the entire manuscript.

3- What did the authors means by “Since the current radiation cooler thickness is in the micron scale, it is difficult to ensure high emissivity and high solar reflectivity balanced with low thermal conductivity characteristics”?. I would suggest to highlight the significance of the work clearly.

4- The outcome of the present work should be compared with the up to date literature.

Reviewer 2 Report

Review comments on polymers-2079596

In this work, a “heat pump film (HPF)” scheme was proposed and a two-layer material was developed for cooling the covered object. HPF, which made of combined ZnO film and PE bubble film via a simple scalable production process, showed high solar reflection and infrared transmis sion characteristics. This manuscript will be considered for acceptable after revised based on the following comments:

1.      In the current state, there are some typographical errors. Therefore, the authors are advised to recheck the whole manuscript for improving the language and structure carefully.

2.      Abstract must be improve with more data. This section needs to be edited

3.      The introduction section is very short and poorly described. It doesn't present the reference to the manuscript scope. In the introduction section, authors should make an in-depth literature review concerning the application of natural and environmentally friendly materials to produce advanced materials that can be used in various fields various fields such as improving human life, energy storage and technologies, and environmental remediation Introduction has deficiency citation to valuable works published before such as: Composites Part B 167 (2019) 643–653; Journal of Colloid and Interface Science 497 (2017) 298–308; Composites Part B 174 (2019) 106930; Journal of Alloys and Compounds 791 (2019) 792-799; Ceramics International 47 (2021) 8959–8972; J Mater Sci: Mater Electron (2022) 33:6549–6554; Polym Eng Sci. 2021;61:2364–2375; ACS Appl Mater Interfaces 2021 Jul 7;13(26):31066-31076; Chemical Methodologies, 4(1), pp. 92-99, https://doi.org/10.33945/SAMI/CHEMM.2020.1.8; Applied Catalysis A, General 621 (2021) 118179. authors should be cite to these works to improve introduction section.

4.      And the structure of the manuscript might need a major adjusting for a better understanding.

5.      Purity of using materials must be clear

6.      The writing logic of characterization analysis is not clear.

7.      Redesign the methods chapter the way so anybody can repeat your procedures.

8.      Results and discussion: - To increase the scientific value of the manuscript Authors should consider extension of the all results section with comparison of obtained results with the results described in previous publications.

9.      The authors should prepare all figures with better resolution

10.  The authors should prepare TEM images for detailed investigation of structure, morphology and size of samples.

11.  Conclusion must be improve with obtaining data and also describe about future

12.  What is the main significance of paper in comparison to other published works?

Round 2

Reviewer 2 Report

The authors have addressed most of my Comments and the manuscript could be acceptable now.